# High-Frequency Near-Infrared Diode Laser Irradiation Attenuates IL-1β-Induced Expression of Inflammatory Cytokines and Matrix Metalloproteinases in Human Primary Chondrocytes

**DOI:** 10.3390/jcm9030881

**Published:** 2020-03-24

**Authors:** Shuzo Sakata, Ryo Kunimatsu, Yuji Tsuka, Ayaka Nakatani, Tomoka Hiraki, Hidemi Gunji, Naoto Hirose, Makoto Yanoshita, Nurul Aisyah Rizky Putranti, Kotaro Tanimoto

**Affiliations:** Department of Orthodontics and Craniofacial Developmental Biology, Hiroshima University Graduate School of Biomedical Sciences, 1-2-3 Kasumi, Minami-ku, Hiroshima 734-8553, Japan; shuzosakata@hiroshima-u.ac.jp (S.S.); tsuka1@hiroshima-u.ac.jp (Y.T.); anakatan@hiroshima-u.ac.jp (A.N.); tomoka1012@hiroshima-u.ac.jp (T.H.); macchanodoame.guhi@gmail.com (H.G.); hirose@hiroshima-u.ac.jp (N.H.); m-yanoshita@hiroshima-u.ac.jp (M.Y.); d196538@hiroshima-u.ac.jp (N.A.R.P.); tkotaro@hiroshima-u.ac.jp (K.T.)

**Keywords:** high-frequency near-infrared diode laser, osteoarthritis, inflammation, matrix metalloproteinase, human chondrocyte

## Abstract

High-frequency near-infrared diode laser provides a high-peak output, low-heat accumulation, and efficient biostimulation. Although these characteristics are considered suitable for osteoarthritis (OA) treatment, the effect of high-frequency near-infrared diode laser irradiation in in vitro or in vivo OA models has not yet been reported. Therefore, we aimed to assess the biological effects of high-frequency near-infrared diode laser irradiation on IL-1β-induced chondrocyte inflammation in an in vitro OA model. Normal Human Articular Chondrocyte-Knee (NHAC-Kn) cells were stimulated with human recombinant IL-1β and irradiated with a high-frequency near-infrared diode laser (910 nm, 4 or 8 J/cm^2^). The mRNA and protein expression of relevant inflammation- and cartilage destruction-related proteins was analyzed. Interleukin (IL) -1β treatment significantly increased the mRNA levels of IL-1β, IL-6, tumor necrosis factor (TNF) -α, matrix metalloproteinases (MMP) -1, MMP-3, and MMP-13. High-frequency near-infrared diode laser irradiation significantly reduced the IL-1β-induced expression of IL-1β, IL-6, TNF-α, MMP-1, and MMP-3. Similarly, high-frequency near-infrared diode laser irradiation decreased the IL-1β-induced increase in protein expression and secreted levels of MMP-1 and MMP-3. These results highlight the therapeutic potential of high-frequency near-infrared diode laser irradiation in OA.

## 1. Introduction

Osteoarthritis (OA) is well-known articular joint disease often resulting in joint pain and disability in adults. Worldwide, 10% of people older than 60 years are estimated to manifest OA symptoms [1]. Furthermore, OA has been shown to be associated with aging, being overweight or obese, cartilage injuries, and genetic factors [2,3,4]. Mechanical stress-induced inflammation plays an important role in OA. Synovitis in the articular condylar surface followed by secretion of inflammatory cytokines such as interleukin 1β (IL-1β) from synovial cells is considered to be the initial stage of OA [5]. The secreted IL-1β infiltrates the cartilage and synovial fluid, causing chondrocyte inflammation [6]. The next stage of OA involves cartilage degradation, particularly extracellular matrix (ECM) destruction by matrix metalloproteinases (MMPs) secreted from inflammatory chondrocytes [7]. IL-1β, tumor necrosis factor alpha (TNF-α), and MMPs are considered catabolic factors in the erosion and proteolysis of cartilage ECM components, including collagen type II and aggrecan [8,9]. Therefore, current studies focus on the coordination, upregulation, or downregulation of these anabolic and catabolic factors for the development of new therapeutic approaches for OA [10,11]. Decreased joint lubrication is also involved in initial OA stages. We previously investigated the mechanism involved in OA and reported that the protein lubricant superficial zone protein (SZP) localizes on the temporomandibular joint surface [12]. We also demonstrated that excessive mechanical loading lowers SZP production [13]; increases the friction coefficient; increases MMP-1, 3, and 9 production; and destroys substrates such as type II collagens, aggrecan, and hyaluronic acid [14,15].

Standard treatments for knee OA include oral nonsteroidal antiinflammatory drugs (NSAIDs), intra-articular corticosteroid or hyaluronic acid injection, and surgery [16,17,18]; however, the international clinical guidelines lack consensus. To identify novel OA therapeutic approaches, we previously investigated the possibility of and mechanism underlying OA inflammation reduction by drug administration. Celecoxib, a selective COX-2 inhibitor, is known to exert protective effects on ECM metabolism in mandibular condylar chondrocytes under excessive mechanical stress [19]. Decactinib, a FAK inhibitor, and semaphorin 3A were shown to inhibit inflammation in chondrocytes under cyclic tensile strain (CTS) [20,21]. However, allopathic drugs cause adverse effects such as upper gastrointestinal complications [22], myocardial infarction, and renal insufficiency [23], whereas joint injection and surgery involve tissue injury risk. Therefore, physical therapies such as ultrasound [24], electrical stimulation [25], exercise [26], spa therapy [27], and low-level laser therapy (LLLT) have been introduced. LLLT is listed as a nonpharmacological and noninvasive treatment option for OA [28], and intensive LLLT offers clinically relevant short-term pain relief for knee OA [29,30,31].

However, laser irradiation induces photothermal effects such as increased tissue activities due to a moderate increase in heat. Therefore, excessive heat damages the surface of tissues [32]. The achievement of clinically desired outcomes with LLLT depends on parameters such as wavelength, pulse frequency, peak power, and time [33]. In general, laser devices generate continuous waves (CWs) or pulse waves (PWs) [33]. LLLT is beneficial when CWs are used, although PWs have different biological and clinical effects [34,35]. Use of the PW mode may be preferred over the CW mode because the “off” times are longer than the “on” times, leading to less heating of cells and possibly permitting much higher peak power densities [32,36,37]. The more favorable effects of PW may be due to the use of a fundamental frequency that is present in biological systems, which is tens to hundreds of hertz. On the other hand, PW may be favorable because some biological processes occur on a time scale of a few milliseconds [38]. LLLT in PW mode better penetrates through melanin and other features of the skin. Thus, pulsing may be the best strategy to reach deep tissues and organs [38]. Furthermore, a comparison of PW and CW near-infrared lasers demonstrated that the wound healing efficacy of the PW mode is more promising than that of the CW mode in vivo [39]. In addition, short-pulse lasers exert not only a biostimulative effect but also a mechanical stimulative effect [40], suggesting that they may have a higher activation effect than CW lasers [39].

Recently, high-frequency near-infrared diode laser devices, including short-pulse lasers (nano-second order), have been developed [32,36]. These devices produce low-level output, high peak power, and no thermal damage to the treated tissue and enable efficient light penetration into the tissue [32,36]. High-frequency near-infrared diode laser irradiation enhanced the proliferation and migration of human gingival epithelial cells [41] and mouse calvarial osteoblasts [42]; promoted epithelialization and bone formation in tooth extraction sockets, possibly by activating cell proliferation and differentiation in vivo [43]; and reduced temporomandibular joint pain [44,45]. Moreover, high-frequency near-infrared diode laser irradiation of periodontal tissue leads to metabolic activation, thereby increasing the tooth movement rate in rats [46]. Thus, high-frequency near-infrared diode laser irradiation may affect chondrocyte metabolism. Compared to LLLT, high-frequency near-infrared diode laser irradiation is highly efficient, enables access to deeper tissues owing to the high intensity of the laser [18], and generates high peak output power. Furthermore, it does not cause excessive heat accumulation in the target tissue, thus overcoming an important concern for clinicians using laser therapies. Therefore, high-frequency near-infrared diode laser therapy may be considered a novel, effective, and safe therapy for OA. However, the effect of high-frequency near-infrared diode laser irradiation on OA cartilage has not yet been reported. Therefore, in this study, we investigated the effect of high-frequency near-infrared diode laser irradiation on the expression of inflammatory cytokines and MMPs in an in vitro OA model and attempted to highlight the potential of such laser devices in OA treatment.

## 2. Materials and Methods

### 2.1. Cell Culture

Normal human articular chondrocyte-knee (NHAC-Kn) cell line was obtained from Lonza. NHAC-Kn cells (CC-2550; Lonza, Walkersville, MD, USA) were cultured in chondrocyte basal medium (Chondrocyte Growth Medium BulletKit^TM^; CC-3217; Lonza) supplemented with 10% fetal bovine serum (FBS; CC-3217; Lonza), chondrocyte growth factors (0.2% R3-IGF-1, 0.5% hrFGF-β, 0.1% transferrin, 0.2% insulin) (CC-3217; Lonza), and 0.1% gentamycin/amphotericin B (CC-3217; Lonza) in an atmosphere containing 5% CO_2_. The culture medium was changed every other day until the cells reached 80% confluence. The cells were then detached from the dish surface using trypsin/EDTA solution (Chondrocyte ReagentPack^TM^; CC-3233; Lonza) and transferred to new dishes. The cells were subcultured according to the manufacturer’s instructions, and passage 5 cells were used for all the experiments.

### 2.2. Laser Irradiation

A high-frequency pulsed laser with 910 nm wavelength, 45 W maximum output power, 300 mW average output power, 200 ns pulse width, 30 kHz pulse repetition rate, and 9.6 cm^2^ irradiation area (Lumix 2; USA Laser Biotech, Inc., Richmond, VA, USA) was used for the experiment. The depth of the medium in each culture dish was adjusted to 2 mm, and the laser was fixed at a height of 40 mm so that cells in one well of a 6-well plate at 1 degree. The following laser presets were used: program 1; pulse rate, 30 kHz; and overall duty cycle, 0.6%.

The output was monitored using a LabMax-TOP laser power meter (Coherent NA, Inc., Wilsonville, OR, USA) connected to a PM10 power sensor (Coherent NA, Inc.). The equipment also emits 650 nm light as the guiding wavelength. Table 1 shows the physical parameters used in the laser irradiation experiments.

NHAC-Kn cells were seeded (1 × 10^5^ cells/well) in 6-well plates (Becton Dickinson, Franklin Lakes, NJ, USA) and cultured in chondrocyte basal medium supplemented with 10% FBS, chondrocyte growth factors (0.2% R3-IGF-1, 0.5% hrFGF-β, 0.1% transferrin, 0.2% insulin), and 0.1% gentamycin/amphotericin B. At 80% confluence; the cells were starved in serum- and growth factor-free chondrocyte basal medium for 24 h. The cells were treated with 10 pg/mL recombinant human IL-1β (PeproTech, Rocky Hill, NJ, USA) for 4 h (for gene expression analysis) or 20 h (for protein expression analysis); irradiated at 0, 4, or 8 J/cm^2^; and cultured for 12 h without FBS.

### 2.3. Quantitative Real-Time Polymerase Chain Reaction (PCR) Analysis

The mRNA levels of IL-1β, IL-6, TNF-α, MMP-1, MMP-3, MMP-9, and MMP-13 were determined by quantitative real-time PCR analysis using a LightCycler system (Roche Diagnostics, Basel, Switzerland) and QuantiTect SYBR Green PCR Master Mix (Qiagen, Tokyo, Japan). Briefly, total RNA was extracted from cultured NHAC-Kn cells using an RNeasy Mini Kit and reverse transcribed using the ReverTra Ace Kit (Toyobo, Osaka, Japan). Real-time PCR was performed using the Thunderbird STBR qPCR Mix (Toyobo) with specific primer sets (Table 1). The relative mRNA levels were analyzed using the ΔΔCt method and normalized to beta actin (ACTB) mRNA levels.

### 2.4. Two-Color Western Blot Analysis

Total protein was extracted from cultured NHAC-Kn cells using RIPA lysis buffer (Nacalai Tesque, Kyoto, Japan) supplemented with 1% (v/v) protease inhibitor cocktail. The cell lysates were centrifuged at 4 °C at 15,000 *g* for 20 min, and protein concentrations in the supernatant were measured using the bicinchoninic acid assay (BCA Protein Assay Kit, Thermo Fisher Scientific, Rockford, IL, USA). Equal amounts of protein were electrophoresed on 10% polyacrylamide gels (e-PAGEL, ATTO, Tokyo, Japan) and transferred to PVDF membranes using the iBlot 2 Gel Transfer Device and iBlot 2 Transfer Stacks (Thermo Fisher Scientific), according to the manufacturer’s instructions. Membranes were then blocked with 5% skim milk for 30 min at room temperature. The membranes were incubated overnight at 4 °C with primary antibodies against MMP-1 (Abcam, Cambridge, MA, USA; Cat# ab139332), MMP-3 (Abcam; Cat# ab52915), and β-actin (Wako, Osaka, Japan), and then incubated with Alexa Fluor 680-labeled antirabbit secondary antibody (LI-COR, Lincoln, NE, USA) or IRDye 800-labeled antimouse secondary antibody (LI-COR). Protein bands were detected using the Odyssey^®^ infrared imaging system (LI-COR); band intensities were quantitated using the ImageJ software and normalized to β-actin band intensity.

### 2.5. Enzyme-Linked Immunosorbent Assay (ELISA)

Here, 80% confluent NHAC-Kn cells were cultured in serum- and growth factor-free chondrocyte basal medium for 24 h and then treated with IL-1β for 12 h. Then, the cells were irradiated at 8 J/cm^2^ and cultured for 12 h, and the culture supernatants were collected and stored at −80 °C until analysis. Levels of secreted MMP-1 and MMP-3 were measured using the MMP-1 (AB215083, Abcam) and MMP-3 (BMS2014/3, Thermo Fisher Scientific) ELISA kits, respectively, following the manufacturers’ recommendations, followed by measurement of the absorbance at 570 nm using a microplate reader.

### 2.6. Statistical Analysis

Data are represented as mean ± SEM. Groups were compared using one-way ANOVA followed by Dunnett’s post hoc test; *P* < 0.05 was considered to indicate statistical significance.

## 3. Results

### 3.1. Effect of High-Frequency Near-Infrared Diode Laser Irradiation on Gene Expression of Inflammatory Cytokines in NHAC-Kn Cells

qPCR analysis showed that mRNA expression levels of IL-1β, IL-6, and TNF-α were significantly upregulated in NHAC-Kn cells treated with IL-1β for 4 h. However, the mRNA levels of IL-1β, IL-6, and TNF-α in the IL-1β-stimulated chondrocytes laser irradiated at 4 and 8 J/cm^2^ were significantly lower than those in all the other cell groups. These results indicate that laser irradiation significantly attenuated the IL-1β-induced upregulation of these inflammatory cytokines (Figure 1).

### 3.2. Effect of High-Frequency Near-Infrared Diode Laser Irradiation on Gene Expression of MMPs in NHAC-Kn Cells

To evaluate the effect of high-frequency near-infrared diode laser irradiation on the expression of genes associated with cartilage destruction, we investigated MMP-1, MMP-3, MMP-9, and MMP-13 mRNA levels in irradiated NHAC-Kn cells. The MMP-1, MMP-3, and MMP-13 mRNA levels were significantly increased by IL-1β stimulation (Figure 2). The MMP-1 and MMP-3 mRNA levels in IL-1β-stimulated cells irradiated at 4 or 8 J/cm^2^ were lower than those in IL-1β-stimulated cells not exposed to radiation (Figure 2A,B); such an effect was not observed for MMP-13 (Figure 2D). No statistically significant changes were observed in the MMP-9 mRNA levels of NHAC-Kn cells (Figure 2C).

### 3.3. Effect of High-Frequency Near-Infrared Diode Laser Irradiation on Protein Expression of MMPs in NHAC-Kn Cells

Next, we measured the protein expression of MMP-1 and MMP-3 using western blot analysis. The protein expression of MMP-1 and MMP-3 in the IL-1β-stimulated chondrocytes was significantly higher than that in the control cells. Similar to the qPCR results, protein expression of MMP-1 and MMP-3 in IL-1β-stimulated cells irradiated at 8 J/cm^2^ was significantly lower than that in the nonirradiated IL-1β-stimulated cells (Figure 3). In addition, we performed ELISA to evaluate the levels of MMP-1 and MMP-3 secreted in the cell culture supernatant. Secreted MMP-1 and MMP-3 levels in the IL-1β group were considerably higher than those in the control group (Figure 4). However, secreted MMP-1 and MMP-3 levels of the IL-1β-stimulated cells irradiated at 8 J/cm^2^ were significantly lower than those of the nonirradiated IL-1β-stimulated cells (Figure 4).

## 4. Discussion

Recently, LLLT was shown to reduce inflammatory reactions both in vitro and in vivo by regulating proinflammatory gene expression [47]. LLLT (904 nm) diminished inflammatory cell migration in a dose-dependent manner in a mouse model of lipopolysaccharide (LPS)-induced periodontitis, with 3 J/cm^2^ being the most effective dose [48]. Furthermore, LLLT at 780 nm (7.7 J/cm^2^) suppressed IL-6 expression in a rat model of collagenase-induced tendinitis [49]; LLLT at 660 nm significantly reduced IL-6 and TNF-α expression in a rat model of carrageenan-induced pleurisy [50]; and LLLT at 660 nm (8 J/cm^2^) reduced the expression of inflammatory markers such as IL-1β, IL-6, and TNF-α in LPS-stimulated human adipose-derived stem cells [51]. However, the precise mechanism underlying the effect of high-frequency near-infrared diode laser irradiation (904 nm) on cartilage inflammation remains unknown. In this study, we investigated the effect of high-frequency near-infrared diode laser irradiation on cultured human chondrocytes (NHAC-Kn cells). To mimic the degenerative effect of osteoarthritic cartilage, NHAC-Kn cells were stimulated using human recombinant IL-1β [52,53]. IL-1β is a proinflammatory cytokine that interferes with ECM turnover by accelerating cartilage matrix degradation and inducing chondrocyte apoptosis.

In this study, qPCR analysis revealed that IL-1β significantly upregulated IL-1β, IL-6, and TNF-α mRNA levels in NHAC-Kn cells and that these levels were significantly attenuated after irradiation of the IL-1β-stimulated chondrocytes with a high-frequency near-infrared diode laser at 4 or 8 J/cm^2^. The gene expression of IL-1β, IL-6, and TNF-α in rat models of acute joint inflammation [54,55] and in the cartilage of rat models of OA [56] decreased significantly after LLLT (808 nm, 142 J/cm^2^) and laser irradiation (808 nm, 71 J/cm^2^), respectively. Moreover, photobiomodulation therapy (830 nm, 214 J/cm^2^) effectively reduced the OA-induced IL-1β, IL-6, and TNF-α protein expression in experimental rat OA models [57]. Our results are consistent with these previous reports; however, the wavelengths, power numbers, frequency, and pulse numbers of laser irradiation differ.

MMPs are Zn^+2^- or Ca^+2^-dependent endopeptidases that degrade ECM components. The MMP family comprises 24 members, 23 of which are found in mammals [58,59]. MMPs play a key role in pathological conditions such as inflammation, rheumatoid arthritis, OA, cardiovascular diseases, fibrosis, and cancer [59]. As an array of proteases, MMP-1, MMP-3, and MMP-13 degrade type II collagen and proteoglycans, causing cartilage proteolysis [60]. High concentrations of MMP-1 (collagenase-1), which is mainly produced by synoviocytes, were found in the synovial fluid of patients with joint lesions and OA. It degrades aggrecan and collagen types I, II, III, VII, X, and IX [61,62,63,64,65]. MMP-3 (stromelysin-1), which is produced by articular synovial cells, chondrocytes, fibroblasts, and macrophages, has the broadest substrate specificity among MMPs and degrades cartilage proteoglycans; collagen types III, IV, V, VII, and IX; laminin; and fibronectin. It has also been implicated in joint destruction in rheumatoid arthritis. MMP-9 (gelatinase B) is usually secreted by fibroblasts and inflammatory cells, which migrate to lesion sites, and can cleave native type IV and V collagens [66]. The presence of active MMP-9 in the synovial fluid has been reported. Because of limited substrate specificity, MMP-9 does not initiate direct joint destruction. As MMP-9 in synovial fluid is produced by inflammatory cells and platelets, MMP levels are influenced by inflammation and likely reflect joint inflammation [67]. MMP-13 (collagenase- 3) is a member of the collagenase subfamily and is usually found in bone articular cartilage and chondrocytes [68]. MMP-13 was also found in synovial tissue, and has been associated with OA and degradation of cartilage proteoglycans and collagen types I, II, and III [69,70]. Moreover, MMP-13 has been suggested to be involved in pathways associated with MMP-9 activation [58,71]. Therefore, MMP-13 may be considered a critical component of the cellular machinery involved in regulating articular cartilage turnover and a potential therapeutic target for cartilage destruction [68]. Several studies have reported the effect of low-level laser irradiation on MMP synthesis and associated pathological conditions [66,72,73,74,75].

In present study, high-frequency near-infrared diode laser irradiation at 4 and 8 J/cm^2^ reduced MMP-1 and MMP-3 mRNA levels, compared with those in the IL-1β only group. However, the MMP-9 and MMP-13 mRNA levels were not altered by laser irradiation at 4 or 8 J/cm^2^. In addition, we assessed the protein expression and secreted levels of MMP-1 and MMP-3 by western blotting and ELISA, respectively. Laser irradiation at 8 J/cm^2^ significantly reduced the IL-1β-stimulated increase in protein levels of MMP-1 and MMP-3. To our knowledge, this is the first report on the effects of high-frequency near-infrared diode laser irradiation on IL-1β-induced human articular chondrocyte inflammation in an in vitro osteoarthritis model. These results reveal the therapeutic potential of high-frequency near-infrared diode laser irradiation in inflammatory conditions of skeletal elements.

LLLT (830 nm, 1.5 J/cm^2^) was reported to significantly decrease the gene expression of MMP-3 at weeks 6 and 8 and that of MMP-1 and MMP-13 at week 8 in a rabbit model of anterior cruciate ligament transection (ACLT)-induced OA. [76]. Furthermore, LLLT significantly decreased the protein levels of MMP-9 in a rat model of papain-induced cartilage injury [77]. Thus, the MMP-1 and MMP-3 levels observed in the current study are consistent with those reported previously. However, the MMP-9 and MMP-13 levels in this study are not consistent with those in the previous reports. This discrepancy may be attributed to the difference in the method for OA induction. In experimental animal models, OA is induced by ALCT or local papain administration. Our study was an in vitro study involving IL-1β stimulation in human articular chondrocytes. In addition, the power, wavelengths, and pulses of the high-frequency diode laser are different from those of LLLT; therefore, the outcomes of these two laser therapies cannot be compared directly. Histological evaluation of the effects of high-frequency diode lasers using OA animal models is required.

A previous study demonstrated that LLLT suppresses the inflammatory response of human adipose-derived stem cells by increasing intracellular cyclic AMP levels and subsequently decreasing NF-κB activity [51]. LLLT was also shown to suppress the activation of the NF-κB signaling pathway in LPS-treated mesenchymal stem cells by inhibiting phosphorylation of p65 and IκBα, thereby exerting an antiinflammatory effect [78]. It has been reported that the IL-1β-induced increase in MMP-13 expression requires p38 activity, JNK activity, and NF-κB translocation but does not involve MEK, whereas the increase in MMP-1 expression requires p38 and MEK activity but does not involve JNK and NF-κB [79]. MMP-1 and MMP-3 expression in IL-1β-stimulated chondrosarcoma cells was suppressed by p38 and ERK inhibitors but not by JNK inhibitor, and MMP-13 expression was suppressed by p38 and JNK inhibitors but not ERK inhibitor [80]. These findings indicate that the signaling pathway for IL-1β-mediated regulation of MMP-13 is different from that for IL-1β-mediated regulation of MMP-1 and MMP-3 and may not involve MEK/ERK activities. Previous studies indicated that high-frequency diode laser irradiation affected ERK activity but not p38 and JNK activity [41,42]. Our qPCR results show that high-frequency diode laser irradiation decreased the IL-1β-induced MMP-1 and MMP-3 expression but did not affect MMP-13 expression. Collectively, these findings suggest the involvement of the MEK/ERK signaling pathway in the IL-1β- and irradiation-mediated regulation of MMP-1 and MMP-3 expression in chondrocytes.

Because the biological effects of lasers are nonspecific, indirect, transient, and complex, it is difficult to ascertain the precise effect of laser irradiation in cellular signaling pathways. One study reported that helium-neon laser irradiation activated the Akt signaling pathway [81], whereas another study conducted using the same laser reported that laser irradiation inactivated this pathway [82]. Nevertheless, the detailed mechanism underlying the antiinflammatory effect of laser irradiation remains poorly understood. Further studies are necessary to elucidate the precise mechanism underlying the effect of high-frequency diode laser irradiation in chondrocytes.

## 5. Conclusions

The present study demonstrates that high-frequency diode laser irradiation ameliorated inflammation and production of MMP-1 and MMP-3 in IL-1β-stimulated human chondrocytes. Collectively, our findings highlight the potential applicability of high-frequency diode laser irradiation in reducing the inflammation and cartilage destruction observed in OA.

## Figures and Tables

**Figure 1 jcm-09-00881-f001:**
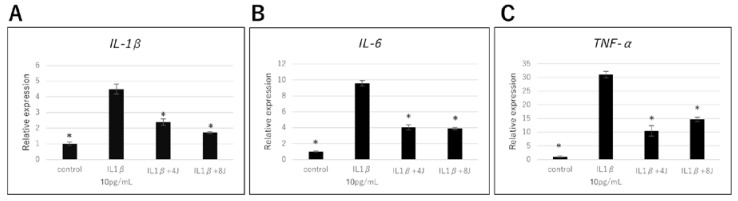
Effect of high-frequency near-infrared diode laser irradiation on interleukin (IL) -1β-induced gene expression of inflammatory cytokines in Normal Human Articular Chondrocyte-Knee (NHAC-Kn) cells. The chondrocytes were treated with IL-1β (10 pg/mL) and laser irradiated (0, 4, 8 J/cm^2^) for 4 h, followed by real-time Polymerase Chain Reaction (PCR) analysis. mRNA expression of (**A**) IL-1β, (**B**) IL-6, and (**C**) tumor necrosis factor (TNF) -α are represented as mean ± SEM of three independent experiments (*n* = 3). * *p* < 0.05 compared to the IL-1β group.

**Figure 2 jcm-09-00881-f002:**
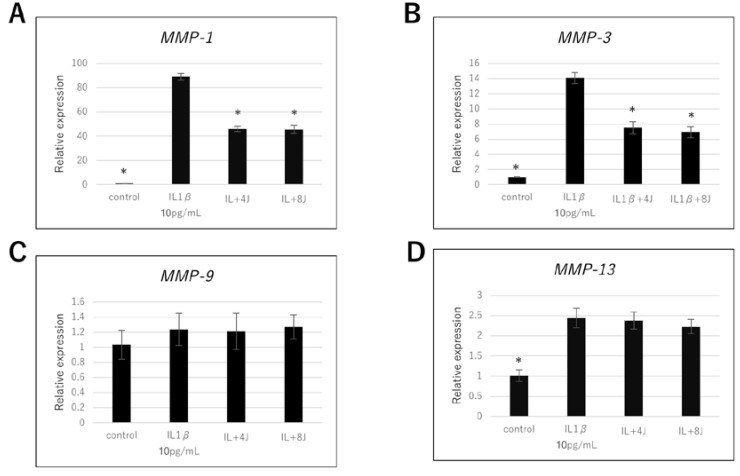
Effect of high-frequency near-infrared diode laser irradiation on interleukin (IL) -1β-induced gene expression of matrix metalloproteinases (MMPs) in NHAC-Kn cells. The chondrocytes were treated with IL-1β (10 pg/mL) and laser irradiated (0, 4, 8 J/cm^2^) for 4 h, followed by real-time polymerase chain reaction (PCR) analysis. mRNA expression of (**A**) MMP-1, (**B**) MMP-3, (**C**) MMP-9, and (**D**) MMP-13 are represented as mean ± SEM of three independent experiments (*n* = 3). * *p* < 0.05 compared to the IL-1β group.

**Figure 3 jcm-09-00881-f003:**
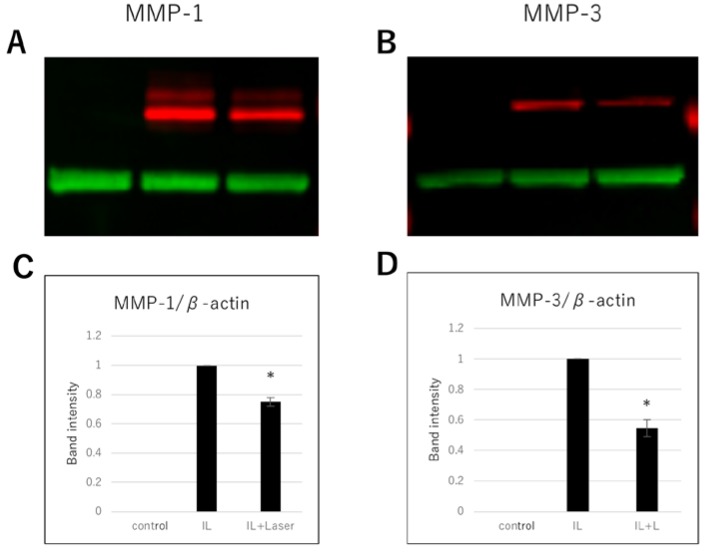
Effect of high-frequency near-infrared diode laser irradiation on interleukin (IL) -1β-induced protein expression of matrix metalloproteinases (MMPs) in NHAC-Kn cells. The chondrocytes were treated with IL-1β (10 pg/mL) and laser irradiated (0, 8 J/cm^2^) for 20 h, followed by two-color western blot analysis. Infrared images of the antibody-treated membranes for (**A**) MMP-1 (red) and (**B**) MMP-3 (red). Band intensities of MMP-1 and MMP-3 were semiquantified using the β-actin (green) band intensity. Relative expression levels of (**C**) MMP-1 and (**D**) MMP-3 normalized to the corresponding levels in the IL-1β group and represented as mean ± SEM of three independent experiments (*n* = 3). * *p* < 0.05 compared to the IL-1β group.

**Figure 4 jcm-09-00881-f004:**
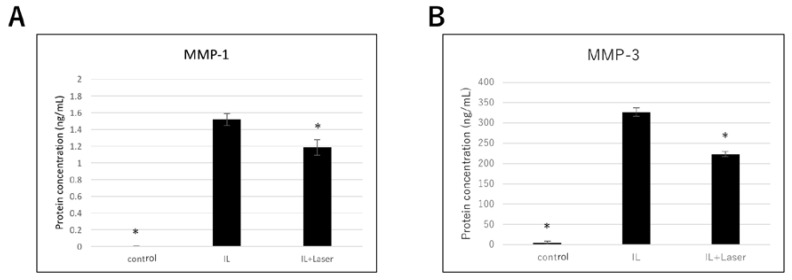
Effect of high-frequency near-infrared diode laser irradiation on levels of secreted matrix metalloproteinases (MMPs) in NHAC-Kn cells. The chondrocytes were treated with interleukin (IL) -1β (10 pg/mL) and laser irradiated (0, 8 J/cm^2^) for 20 h; the culture medium was collected and used for enzyme-linked immunosorbent assay (ELISA). Secreted protein levels of (**A**) MMP-1 and (**B**) MMP-3 are represented as mean ± SEM of three independent experiments (*n* = 3). * *p* < 0.05 compared to the IL-1β group.

**Table 1 jcm-09-00881-t001:** Physical parameters of the laser irradiation.

Parameter	(Unit)	Value
Wavelength	(nm)	910
Operating mode		pulsed
Pulse duration	(ns)	200
Frequency	(kHz)	30
Duty cycle		0.6
Peak power	(W)	45
Average power	(W)	0.3
Average power density	(W/cm^2^)	0.6
Peak power density	(W/cm^2^)	90

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
