# Peer review of "High-Frequency Near-Infrared Diode Laser Irradiation Attenuates IL-1β-Induced Expression of Inflammatory Cytokines and Matrix Metalloproteinases in Human Primary Chondrocytes"

_jcm, 2020, doi:10.3390/jcm9030881_

Round 1

Reviewer 1 Report

The study was undertaken to evaluate the effects of high-frequency near-infrared diode laser irradiation on IL-1β-induced chondrocyte inflammation in an in vitro osteoarthritis model, and drew the conclusion of its effect  of the  the therapeutic potential of high-frequency near-infrared diode laser in inflammatory condition of skeletal elements. The study highlights the effect of the 'photobiomodulation', thereby the idea of the study may not appear to be novel or breaking new ground, the experimental design represents no flaw and also the results and conclusion demonstrates the fine reasoning flow. 

Authors responses

We appreciate your thorough review of our manuscript and your positive comments. To our knowledge, this is the first report on the effects of high-frequency near-infrared diode laser irradiation on IL-1β-induced human articular chondrocyte inflammation in an in vitro osteoarthritis model. This study revealed the therapeutic potential of high-frequency near-infrared diode laser irradiation in inflammatory conditions of skeletal elements. To clarify the novelty of the study, the Discussion section have been revised (page 8, lines 279-283).

Reviewer 2 Report

I'm proud to receive this paper. Very high quality and innovative research.

I need to know T-on Time or T-on% compared to duty cycle.

 And, in order to determine the proper fluency, the distance from the laser beam and the cells, and the time application.

 As i read, Peak power is 45 W, Average power is 300 mW, duty cycle is 200 ns, 30 KHz is the frequency and the beam is 9.6 cm2 irradiation area.

I totally agree on using high frequency and high energy peak power in order to penetrate deeper. The thermal control is perfectly made by a short duty cycle.

Authors responses

We are grateful for your suggestions regarding ways to improve our manuscript, and we appreciate your interest in our research. We have included the physical parameters of the laser irradiation in Table 1 and described the experimental conditions in more detail in Materials and Methods (page 3, lines 114-121). To more clearly explain continuous-wave and pulse-wave lasers, the Introduction section has been revised (page 2, lines 64-80).

Round 2

Reviewer 1 Report

The study was aimed to evaluate the biological effects of high-frequency near-infrared diode laser irradiation on IL-1β-induced chondrocyte inflammation in an in vitro osteoarthritis(OA) model, since he effect of high-frequency near-infrared diode laser irradiation in in vitro/vivo OA models appears rare in spite of the fact that High-frequency laser photobiomodulation provides efficient biostimulation. which is believed to be effective in treatment of osteoarthritis.

The manuscript appears sound methodology and well documented reasoning to draw the conclusion of therapeutic potential of the of high-frequency near-infrared diode laser irradiation to OA model.

Reviewer 2 Report

Especially in this period, when researchers are founding solutions in order to reduce the activity of IL-6 or cytokines  against Covid-19, I think that a paper like this could help the use of LlLLT in all medical fields.
So I agree to review the paper and I consider the results of this research very interesting and well done.